# Tumor Site-Specific Cleavage Improves the Antitumor Efficacy of Antibody–Drug Conjugates

**DOI:** 10.3390/ijms241311011

**Published:** 2023-07-02

**Authors:** Keyuan Xu, Jiani Han, Liu Yang, Li Cao, Shuang Li, Zhangyong Hong

**Affiliations:** State Key Laboratory of Medicinal Chemical Biology, College of Life Sciences, Nankai University, Tianjin 300071, China; 1120190518@mail.nankai.edu.cn (K.X.); 1120200525@mail.nankai.edu.cn (J.H.); 2120211264@mail.nankai.edu.cn (L.Y.); 1120150370@mail.nankai.edu.cn (L.C.)

**Keywords:** HER2, affibody, antibody–drug conjugates (ADCs), MMAE, half-life

## Abstract

Antibody—drug conjugates (ADCs) play important roles in tumor therapy. However, traditional ADCs are limited by the extremely large molecular weight of the antibody molecules, which results in low permeability into solid tumors. The use of small ADCs may be expected to alleviate this problem, but this switch brings the new limitation of a greatly shortened blood circulation half-life. Here, we propose a new cleavable ADC design with excellent tumor tissue permeability and a long circulation half-life by fusing the small ADC Z_HER2_-MMAE with the Fc domain of the antibody for circulation half-life extension, and inserting a digestion sequence between them to release the small ADC inside tumors for better tumor penetration. The experimental results showed that the designed molecule Fc-U-Z_HER2_-MMAE has a significantly increased blood circulation half-life (7.1 h, 59-fold longer) compared to the small ADC Z_HER2_-MMAE, and significantly improved drug accumulation ability at tumor sites compared to the conventional full-length antibody-coupled ADC Herceptin-MMAE. These combined effects led to Fc-U-Z_HER2_-MMAE having significantly enhanced tumor treatment ability, as shown in mouse models of NCI-N87 gastric cancer and SK-OV-3 ovarian cancer, where Fc-U-Z_HER2_-MMAE treatment achieved complete regression of tumors in all or a portion of animals with no obvious side effects and an MTD exceeding 90 mg/kg. These data demonstrate the therapeutic advantages of this cleavable ADC strategy, which could provide a new approach for ADC design.

## 1. Introduction

Antibody–drug conjugates (ADCs) are highly effective compounds for tumor treatment. They are composed of small, highly cytotoxic molecular drugs attached to monoclonal antibodies, which endows the therapeutic molecules with both the high selectivity of antibodies and high cytotoxicity of the small molecule [1]. At present, more than 14 ADCs have been successfully approved for clinical use [2,3], which has significantly improved the survival rates and prognoses of patients. However, conventional ADCs that use large monoclonal antibodies (~150 kDa) as tumor-targeting ligands may suffer from low penetration efficiency into solid tumor tissues due to their large molecular size [4], and thus, using smaller targeting ligands to prepare ADCs is expected to alleviate this problem. To this end, various small antibody fragments, such as antigen-binding fragments (Fabs, ~55 kDa) [5,6], single-chain antibody variable fragments (scFvs, ~28 kDa) [7] and nanobodies (VHHs, ~15 kDa) [8], or miniaturized antibody analogs, such as the designed ankyrin repeat class of scaffold proteins (DARPins, ~18 kDa) [9,10] and affibodies (~6.5 kDa) [11,12] have been widely tested as drug carriers for ADCs. However, these designs have brought new problems, including an ultrashort circulation half-life that is only a few minutes to a dozen minutes, in contrast to the half-life of full-length antibody molecules (15–30 days) [13,14,15], which significantly reduces the therapeutic effect of the conjugated drug molecules.

Fusing drug molecules with the fragment crystallizable (Fc) domain of antibodies or modifying them with polyethylene glycol (PEG) can significantly improve their circulation half-life [13,16,17,18,19,20,21]. However, this strategy will significantly increase the molecular weight of the drug, which in turn will lead to a reduction in its permeability into tumor tissues. The development of a new strategy that can design drugs that possess a long blood circulation half-life and better tumor tissue permeability simultaneously may have important application potential for improving the therapeutic effects of ADCs. In this study, we designed a cleavable ADC molecule that can be specifically cleaved by enzymes in tumor tissue to achieve the above expectations. We fused the Fc domain of the antibody to the small ligand-coupled ADC molecule and added the degradation sequence of a tumor tissue-specific protease between them. This molecule achieved a long circulation half-life in the blood through Fc fusion while restoring its good permeability into tumors after entry into the tumor tissue, where it was cleaved by the specific protease to release the small ADC molecule. A large number of studies have shown that many proteases, such as urokinase-type plasminogen activator (uPA) and matrix metalloproteinase (MMP) [22,23,24], are upregulated in various tumors, including breast cancer, gastric cancer, ovarian cancer, colorectal cancer and other types of tumors [25], which helps to promote the growth and metastasis of tumor cells. Additionally, the overexpression of these proteases is also closely related to the malignancy of tumors [26]. These proteases can be thus used to specifically cleave Fc-fused ADC molecules.

This study aimed to explore the feasibility and effectiveness of the abovementioned strategy (Figure 1A). We selected the affibody Z_HER2:342_ with a K_D_ (equilibrium dissociation constant) of 22 pM to the human epidermal growth factor receptor-2 (HER2) [27] as the tumor-targeting ligand and prepared the small ADC molecule Z_HER2_-MMAE through a site-directed coupling strategy (Figure 1B) catalyzed by sortase A with the microtubule antagonist monomethyl auristatin E (MMAE). Affibodies have a small molecular weight (6–7 kDa) that is only one twentieth that of traditional antibody molecules [28,29,30], so the prepared ADC molecule Z_HER2_-MMAE has a relatively small molecular size. Z_HER2:342_ was obtained through affinity maturation screening based on the first-generation anti-HER2 affibody Z_HER2:4_, and it had higher affinity and better overall tumor uptake than Z_HER2:4_ [27]. MMAE is a highly potent toxin molecule derived from marine organisms, commonly used in the construction of ADC molecules. Among the 14 approved ADC drugs, six of them use MMAE as the payload [31]. Site-specific coupling could potentially improve the chemistry, manufacturing and control (CMC) of ADCs, and thus improve the therapeutic index, which is important for the clinical application of ADC molecules [32,33,34]. For these reasons, many site-specific coupling strategies have been developed [35]. Among them, the one based on the transpeptidase-sortase-A-mediated enzymatic coupling has the advantages of easy operation, high selectivity and robust linkage [35], which was then adopted by us for the preparing the target molecules (Figure 1B). We fused the Fc region of the human antibody to the N-terminus of Z_HER2_-MMAE and inserted the uPA enzyme recognition motif (LSGRSDNH) between them. The prepared molecule was named Fc-U-Z_HER2_-MMAE. We also prepared the conventional ADC molecule Herceptin-MMAE as the control, in which MMAE was site-specifically coupled to the C-terminus of the full-length anti-HER2 antibody Herceptin.

Here, we report in detail the preparation and characterization of this cleavable ADC molecule Fc-U-Z_HER2_-MMAE. This ADC was compared with the small Z_HER2_-MMAE and the conventional full-length antibody-coupled Herceptin-MMAE in terms of activity, in vivo distribution, and tumor treatment ability to provide a new design strategy for the development of efficient ADCs.

## 2. Results and Discussion

### 2.1. Design and Expression of the Raw Proteins

We fused the human IgG1 Fc sequence to the N-terminus of Z_HER2:342_ and inserted a uPA enzyme recognition sequence flanked by a flexible Gly/Ser-rich peptide linker (N-GGGGSGGGGSLSGRSDNHGGGGS-C; the uPA substrate region is underlined) [22,36,37,38] between Fc and Z_HER2:342_ to construct the protein Fc-U-Z_HER2_. The coding sequence of Z_HER2:342_ is based on the amino acid sequence described by Orlova et al. [27]. Moreover, the C-terminus of Fc-U-Z_HER2_ was fused with a sortase A recognition motif (LPETGG) and 6 × His tag for enzyme-catalyzed drug coupling. After sortase A-mediated coupling, the GG-His_6_ in the LPETGG-His_6_ tag was replaced by GGG-VC-PAB-MMAE. The enzyme uPA is highly expressed in breast cancer, gastric cancer, ovarian cancer and other types of tumor tissues, but displays low expression or low activity in normal tissues, which ensures that enzyme digestion is mainly carried out at tumor sites. Additionally, we constructed expression plasmids for the full-length antibody Herceptin and small molecule ligand Z_HER2_, which were used to prepare the control conjugates Herceptin-MMAE and Z_HER2_-MMAE. For these two proteins, the sortase A recognition motif and 6 × His tag were also fused at the C-terminus for enzyme-catalyzed drug coupling. For the small Z_HER2_ molecule, another 6 × His-tag was added to the N-terminus for the convenience of downstream purification of the conjugate Z_HER2_-MMAE. The drug-to-antibody ratio (DAR) of Fc-U-Z_HER2_-MMAE and Herceptin-MMAE was 2, while the one of Z_HER2_-MMAE was 1.

Fc-U-Z_HER2_ and Herceptin were transiently expressed in mammalian human embryonic kidney cells (HEK-293F) and purified by a protein A chromatography column, and Z_HER2_ was expressed in BL21 (DE3) pLysS cells followed by purification with a nickel chelate affinity chromatography column. After purification, sodium dodecyl sulphate-polyacrylamide gel electrophoresis (SDS—PAGE) was used to verify the molecular weights and purities of these proteins. As shown in Appendix A, these proteins had the expected molecular weights, and their purities were estimated to be more than 95%. To verify the ability of Fc-U-Z_HER2_ to be cleaved by the uPA enzyme, Fc-U-Z_HER2_ was incubated with human uPA, and the digested products were analyzed by SDS—PAGE. As shown in Appendix A, Fc-U-Z_HER2_ was cleaved into two bands under uPA treatment.

### 2.2. Enzymatic Coupling of MMAE Catalyzed by Sortase A

The preparation of the conjugates was achieved by site-specific coupling catalyzed by sortase A (Figure 1B). Sortase A is a transpeptidase derived from *Staphylococcus aureus*. It can specifically recognize the sequence Leu-Pro-Xxx-Thr-Gly (LPXTG, where X is any amino acid and glycine cannot be a free carboxylate) [35] and efficiently couple nucleophilic reagents with multiple glycine residues at the N-terminus to the threonine site of LPXTG [39] via an acyl-enzyme intermediate formed from the active site cysteine residue of soratse A and the threonine residue of the LPXTG tag. Here, the cytotoxic small molecule MMAE with three glycine residues and a suitable linker modified at the end (GGG-VC-PAB-MMAE) was used as the nucleophilic reagent for coupling to the C-terminus of the proteins through sortase A catalysis. Sortase A was expressed in an *Escherichia coli* system with a glutathione S-transferase (GST) tag for purification and analyzed by SDS—PAGE (Appendix A).

The exploration of the sortase A-catalyzed coupling reaction conditions is shown in Appendix A, where the influence of GGG-VC-PAB-MMAE equivalence, sortase A concentration, reaction time and temperature on the reaction efficiency was tested. When the ratio of GGG-VC-PAB-MMAE to Fc-U-Z_HER2_ was 2:1, the newly formed band of the product (Fc-U-Z_HER2_-MMAE, red arrow) can be seen below that of Fc-U-Z_HER2_ (green arrow). When the ratio increased to 10:1, more than 90% of the raw protein was converted into the product (Appendix A). During enzyme-catalyzed coupling, the LPETGG-6 × His tags at the end of Fc-U-Z_HER2_ were replaced with LPETGG-VC-PAB-MMAE, and SDS—PAGE showed different bands. Appendix A shows the influence of the concentration of sortase A on the reaction efficiency. When the final concentration of transpeptidase sortase A was 2.5 μM or above, more than 95% of the raw protein was transformed to the product. The exploration of the reaction time is shown in Appendix A. When the reaction time was less than 2 h, the reaction efficiency was less than 20%, and after extending the reaction time to 8 h or more, 95% of Fc-U-Z_HER2_ was converted into the product. Appendix A shows the influence of the reaction temperature. The reaction can be carried out at 4 °C, 25 °C or 37 °C, but a temperature of 4 °C seemed to be ideal. Based on the above exploration, the following reaction conditions were used for the preparation of the conjugate Fc-U-Z_HER2_-MMAE: 40 μM Fc-U-Z_HER2_, 400 μM GGG-VC-PAB-MMAE, and 2.5 μM sortase A at 4 °C for 8 h. The same reaction conditions were also used for the preparation of the Herceptin-MMAE and Z_HER2_-MMAE conjugates (Appendix A).

The coupling products of Fc-U-Z_HER2_-MMAE and Herceptin-MMAE were purified with a protein A affinity chromatography column, and the coupling product Z_HER2_-MMAE was purified with a nickel chelate affinity chromatography column. The purity of these products was examined by SDS—PAGE (Figure 2A), and the results showed that more than 90% of the raw proteins were successfully linked to MMAE. Western blotting (Figure 2B) was used to further verify the coupling of MMAE to the protein molecules, where mouse anti-human IgG Fc monoclonal antibody, mouse anti-6 ×His monoclonal antibody, or mouse anti-MMAE monoclonal antibody were used as the detection antibodies. These results confirmed that most of the raw proteins had been successfully coupled with MMAE. We further preliminarily tested the stability of the conjugate Fc-U-Z_HER2_-MMAE by SDS—PAGE (Appendix A). The results show that the conjugate has high storage stability at −80 °C and −20 °C without obvious fracture.

### 2.3. Ability of Fc-U-Z_HER2_-MMAE to Bind to the HER2 Receptor

Enzyme-linked immunosorbent assay (ELISA) was used to determine the ability of these conjugates to bind to the HER2 receptor (Figure 2C,D). The human HER2 protein was coated on plates, coincubated with various concentrations of the conjugates, and then incubated with mouse anti-MMAE monoclonal antibody as the primary antibody and horseradish peroxidase (HRP)-labeled goat anti-mouse as the secondary antibody. The results showed that the EC_50_ values of Fc-U-Z_HER2_-MMAE, Z_HER2_-MMAE and Herceptin-MMAE with the HER2 receptor were 0.95 nM, 1.91 nM and 0.04 nM, respectively. Fc-U-Z_HER2_-MMAE appeared to have a binding affinity similar to that of Z_HER2_-MMAE against the HER2 receptor but ten times lower than that of Herceptin-MMAE. Theoretically, maintaining a relatively low binding ability may be beneficial for Fc-U-Z_HER2_-MMAE to reduce the on-target/off-tumor toxicity against normal organs before its cleavage at the tumor site. Some normal organs, such as the heart, also express the HER2 receptor, which results in side toxicity of anti-HER2 ADCs in the clinic. Compared to Z_HER2_-MMAE, Fc-U-Z_HER2_-MMAE did not seem to increase the binding affinity to HER2 receptor due to the Fc bivalent effect, which may be related to the fusion of Fc at the N-terminus of the Z_HER2_ molecule in our design. The N-terminus is an important site at which Z_HER2_ bind to its receptor, and this fusion may cause some steric obstacles for their interaction, thus offsetting the increase in the affinity caused by the Fc bivalent effect. The ELISA results in Appendix A show that Fc-U-Z_HER2_-MMAE still has a strong binding ability with the HER2 receptor after cleavage by uPA.

### 2.4. Subcellular Colocalization of the Conjugates Determined by FACS and Confocal Microscopy

ADCs need to be internalized into the cytoplasm to produce specific cytotoxicity. Therefore, we further examined the binding of these conjugates to cells and their localization inside cells using flow cytometry and confocal microscopy. In these experiments, HER2-overexpressing NCI-N87 tumor cells and HER2-negative H-CC70 cells were incubated with the conjugates at 4 °C for 1 h and then stained with mouse anti-His or anti-MMAE monoclonal antibody and goat-anti-mouse IgG-FITC conjugate before being observed with flow cytometry and microscopy. Flow cytometry (Appendix A) showed that HER2-overexpressing tumor cells including NCI-N87, SK-BR3, BT-474 and SK-OV-3 cells had a strong fluorescence signal after treatment with either of these three conjugates, while HER2-negative H-CC70 cells did not. Confocal microscopy (Appendix A) showed that NCI-N87 cells overexpressing HER2 had notable accumulation of fluorescent dyes in the cytoplasm, while HER2-negative H-CC70 cells did not. Visualization of the intracellular location of these conjugates through lysosomal staining is shown in Appendix A. FITC-labeled conjugates (green) clearly accumulated in LysoTracker-labeled lysosomes (red), which may indicate that these conjugates could be decomposed for drug release through a lysosomal pathway.

### 2.5. Cytotoxicity Analysis of Fc-U-Z_HER2_-MMAE Determined by MTT Assays and FACS

Next, we examined the cytotoxicity of these conjugates using MTT assays and FACS. The MTT assay results (Figure 3A,B) showed that these conjugates had extremely high toxicity to HER2 high-expressing NCI-N87, SK-BR-3 and BT-474 cells, which occurred in a HER2 receptor-dependent manner, while they had reduced toxicity to SK-OV-3 cells that have moderate HER2 receptor expression and minimal toxicity to H-CC70 cells that have low HER2 expression (IC_50_ > 500 nM). Among them, Fc-U-Z_HER2_-MMAE had IC_50_ values of 1.51 nM (NCI-N87), 0.55 nM (SK-BR-3), 0.34 nM (BT-474) and 41.2 nM (SK-OV-3) in these HER2-expressing cells, with IC_50_ values of 0.65 nM (NCI-N87), 0.43 nM (SK-BR-3), 0.87 nM (BT-474) and 56.5 nM (SK-OV-3) for Herceptin-MMAE and 11.1 nM (NCI-N87), 0.73 nM (SK-BR-3), 2.10 nM (BT-474) and 62.5 nM (SK-OV-3) for Z_HER2_-MMAE. When the conjugate Fc-U-Z_HER2_-MMAE was treated with the uPA enzyme, it still showed a strong ability to kill HER2-overexpressing cells. Fluorescence microscopy observations (Appendix A) also showed that the survival of the cells (green) highly expressing HER2 decreased sharply after treatment with these conjugates, which was consistent with the MTT results.

FACS was used to further explore the killing effect of these conjugates on HER2-overexpressing cells through annexin V-APC/PI staining for apoptosis (Appendix A). When SK-BR-3 and H-CC70 cells were treated with 10 nM Fc-U-Z_HER2_-MMAE for 24, 48 and 72 h, the apoptosis rate (including early and late apoptosis) of HER2-positive SK-BR-3 cells increased from 13.1% (24 h) to 36.6% (48 h) and 59.7% (72 h), while HER2-negative H-CC70 cells did not show significant apoptosis after 72 h of treatment. The cells treated with Herceptin-MMAE and Z_HER2_-MMAE showed similar results. These results demonstrate that these conjugates can selectively accelerate the apoptosis of cells highly expressing HER2, which is very consistent with the MTT results. When these cells were preincubated with the Z_HER2_ protein to block the HER2 receptors (Appendix A) and then incubated with the conjugates, the apoptosis rates induced by Fc-U-Z_HER2_-MMAE and Z_HER2_-MMAE were greatly reduced, while those induced by Herceptin-MMAE were unaffected, which verifies that Z_HER2_ has a different receptor binding epitope than Herceptin [40,41]. Flow cytometry was further used to analyze the cell cycle changes after treatment with these conjugates (Appendix A). After the cells were treated with the conjugates and stained with PI, HER2-positive NCI-N87 cells showed 24.80% (Z_HER2_-MMAE), 35.54% (Herceptin-MMAE) and 33.86% (Fc-U-Z_HER2_-MMAE) G2/M phase cells, while HER2-negative H-CC70 cells were almost free of cycle-arrested cells.

### 2.6. Pharmacokinetics and Biodistribution of the Conjugates in Mice

ELISA was used to determine the blood circulation half-life values of Fc-U-Z_HER2_-MMAE and the control conjugates (Figure 4A,B and Appendix A). The terminal half-life (t_1/2_) of Z_HER2_-MMAE is very short, only 0.12 h (7.2 min), and the area under the curve (AUC) is only 186.6 mg/L × h. This is consistent with the short half-life of small ADC molecules reported in the literature [42,43]. Fc-U-Z_HER2_-MMAE had an extended t_1/2_ of approximately 7.1 h, which was 59-fold longer than that of Z_HER2_-MMAE, and its AUC also increased to 1901.0 mg/L × h. Fc-U-Z_HER2_ without MMAE had a t_1/2_ of 7.0 h, which is quite similar to that of Fc-U-Z_HER2_-MMAE, indicating that the coupling of MMAE had no effect on the circulation half-life of the Fc-U-Z_HER2_ protein. However, there remained a certain gap between the half-life of Fc-U-Z_HER2_-MMAE and that of Herceptin-MMAE, which reached 186.9 h. This may have been caused by the presence of uPA cleavage sites in the Fc-U-Z_HER2_-MMAE molecule. However, it remains unknown whether this event was caused by the cleavage of uPA in tumor tissue or by early cleavage during blood circulation.

We further monitored the targeted distribution of these conjugates in mice using an in vivo imaging system. Here, the conjugates were first fluorescently labeled with Cy5 (Figure 4C,D) and then injected into NCI-N87 tumor-bearing nude mice via the tail vein (20 nmol). As shown in Figure 4E, Fc-U-Z_HER2_-MMAE had excellent enrichment at tumor sites. It began to accumulate in the tumor 1 h postinjection and was distributed mainly in the tumor 4 h postinjection; this high distribution in the tumor was maintained between 12 h and 48 h postinjection. Z_HER2_-MMAE also had certain enrichment at the tumor, but its circulation half-life was too short, and the amount of residual drug in the body at 4 h postinjection was minimal. Due to its large molecular weight, Herceptin-MMAE had very limited enrichment in the tumor and did not show high accumulation throughout the monitoring period (1 to 48 h postinjection). Figure 4F shows the fluorescence intensity of the organs after quantitative analysis. The above results confirmed that, compared with high molecular weight Herceptin-MMAE and Z_HER2_-MMAE with a short circulation half-life, the cleavable ADC molecule Fc-U-Z_HER2_-MMAE had greatly improved tumor accumulation ability.

### 2.7. Detection of uPA Enzyme Distribution and Activity

Distribution of the uPA enzyme in tumor tissues and other normal organs in vivo is crucial for the cleavage and activation of Fc-U-Z_HER2_-MMAE. Therefore, we used an ELISA kit to detect the expression of the uPA enzyme in the homogenate supernatants of tumor tissues and normal organs from NCI-N87 tumor model mice and found that the expression of the uPA enzyme in NCI-N87 tumor tissue was significantly higher than that in other tissues (Appendix A). This result was confirmed by immunohistochemistry (IHC) section staining (Appendix A), where there were evident brown antibody signals and orange positive grade map signals in the tumor tissues. These results confirmed that the uPA enzyme was highly expressed in NCI-N87 tumor tissue, but not in most normal organs, including liver, spleen, lungs, kidneys and muscles, except that the heart seems to have a certain degree of uPA expression. However, studies have shown that, due to the presence of endogenous inhibitors in healthy tissues, even if there is a certain expression of uPA in these tissues, its proteolytic activity is highly inhibited [22,36,37]. In addition, the uPA digestion recognition sequence we selected was reversely selected by a wide range of proteases [38], which can further ensure that our substrate is safe in healthy tissues.

We further verified the cleavage performance of the uPA enzyme through an ex vivo experiment by designing an indicator protein Fc-Z_E01_-U-Z_HER2_, in which Z_E01_ is a sequence with a strong interaction with Z_HER2_ that blocks the binding of the latter with the HER2 receptor [44]. When the indicator protein is cleaved by uPA, the released Z_HER2_ binds to the HER2 receptor. Moreover, Fc-Z_E01_-A-Z_HER2_ was prepared as a control, in which the uPA digestion sequence was replaced by a uncleavable alanine linker (expressed as A), so it could not bind to the HER2 receptor even in the presence of uPA. A 6 × His tag was fused to the C-terminus of Z_HER2_ for antibody detection. The two molecules were coincubated with tissue sections from HER2-positive NCI-N87 tumors and mouse hearts and then detected with a Cy3-labeled anti-His antibody (Appendix A). The tumor tissue from the Fc-Z_E01_-U-Z_HER2_ treatment group showed a strong fluorescence signal, while the tumor tissue from the Fc-Z_E01_-A-Z_HER2_ treatment group did not. Neither treatment group showed a fluorescence signal in the mouse hearts, while the groups treated with the control molecules Z_HER2_ and Herceptin showed strong fluorescence signals. These results confirmed that there was high expression of HER2 receptor and high proteolytic activity of uPA enzyme in NCI-N87 tumor tissue, where uPA could effectively cleave the Fc-Z_E01_-U-Z_HER2_ molecule so that the released Z_HER2_ molecule interacted with the HER2 receptor. However, mouse heart tissue overexpresses the HER2 receptor but does not have uPA proteolytic activity. These data may indicate that our design could be expected to reduce the side effects to the heart and other normal tissues that express HER2.

### 2.8. In Vivo Antitumor Efficacy of the Conjugates

Next, we constructed subcutaneous mouse xenograft tumor models to evaluate the in vivo tumor therapeutic ability of Fc-U-Z_HER2_-MMAE. First, we constructed an NCI-N87 gastric tumor model and treated the mice with Fc-U-Z_HER2_-MMAE and the control conjugates at the same molar amount of MMAE (170 nmol/kg MMAE) (Figure 5A,B). For Fc-U-Z_HER2_-MMAE and Herceptin-MMAE, mice were treated once a week for six weeks, and for Z_HER2_-MMAE, which has a short half-life, the frequency of administration increased to twice a week for six weeks. Figure 5A shows the changes in mouse tumor growth after treatment. In the Fc-U-Z_HER2_-MMAE treatment group, tumor growth could be significantly inhibited. Three of the five mice achieved complete tumor regression at the end of the observation period (Day 90), and the other two also showed great inhibition of tumor growth with an average tumor volume of only 50 mm^3^ at Day 90. In the Z_HER2_-MMAE treatment group, although the treatment frequency doubled to twice a week, tumor growth was only partially inhibited, and no mouse tumors had been completely cleared. Herceptin-MMAE showed good inhibition of tumor growth, but its effect was significantly weaker than that of Fc-U-Z_HER2_-MMAE, with none of the mice showing complete tumor regression in this group. Herceptin-MMAE and Z_HER2_-MMAE could not achieve complete tumor removal, while cleavable Fc-U-Z_HER2_-MMAE showed a much stronger ability to treat tumors, with 60% of the mice showing complete tumor removal at a dose of 170 nmol/kg. Throughout the observation period, the body weights of the mice (Figure 5B) in each treatment group did not change significantly, and the survival of the mice was normal, indicating that the mice should have good tolerance to treatment with these conjugates at the administered doses.

We also collected serum from the mice in each group to detect the levels of ALT (alanine aminotransferase), AST (aspartate aminotransferase (AST), BUN (urea/urea nitrogen) and CR (creatinine) (Appendix A) to evaluate the effects of treatment on liver and renal function. The results showed that AST, BUN and CR levels in all treatment groups remained within the normal range, showing no difference from those in the PBS control group, while the ALT levels in the Herceptin-MMAE group seemed to increase slightly. In addition, we collected major organs (heart, liver, spleen, lung and kidney) from these mice and prepared paraffin sections for hematoxylin-eosin (H&E) staining analysis, and these results also showed that there was no apparent damage to these organs in all treatment groups (Appendix A).

Subsequently, we constructed an SK-OV-3 subcutaneous tumor model and used similar doeses of Fc-U-Z_HER2_-MMAE and other control conjugates once a week for a total of six administrations. It can be seen from the tumor growth curves (Figure 5C) that tumor growth was also significantly inhibited by Fc-U-Z_HER2_-MMAE treatment, and tumor volume remained below 50 mm^3^ on Day 30 and below 250 mm^3^ on Day 52. One of the five mice achieved complete tumor regression on Day 90. In the Z_HER2_-MMAE treatment group, even though the frequency of administration doubled, the tumors of the mice still rapidly grew, and the tumor volume increased to more than 500 mm^3^ on Day 52. Herceptin-MMAE showed a good antitumor effect in the SK-OV-3 tumor model, but its effect was significantly weaker than that of Fc-U-Z_HER2_-MMAE, and the mouse tumor volume increased to approximately 480 mm^3^ on Day 52, which was more than 2.2-fold that in the Fc-U-Z_HER2_-MMAE treatment group at the same dose. The body weight of the mice (Figure 5D) in the group of Fc-U-Z_HER2_-MMAE increased appropriately and their physiological behavior was completely normal, which indicates that SK-OV-3 tumor-bearing mice tolerated Fc-U-Z_HER2_-MMAE treatment well at this dose. However, in the groups of Z_HER2_-MMAE, Herceptin-MMAE and the PBS control, the body weights of the mice seemed to show a slight decrease during treatment and some mice also had abnormal physiological conditions, including spinal curvature and wet stools.

Next, we further tested the therapeutic ability of Fc-U-Z_HER2_-MMAE against relatively large tumors. We constructed NCI-N87 and SK-OV-3 tumor models with initial tumor volumes of 500 mm^3^ to 1100 mm^3^ for treatment with Fc-U-Z_HER2_-MMAE. In the NCI-N87 model animals with initial tumor volumes of 500 mm^3^ (Figure 6A,B), 27 mg/kg Fc-U-Z_HER2_-MMAE (corresponding to 340 nmol/kg MMAE) was used for treatment every 7 days with a total of nine doses or every 14 days with a total of five doses. In the group that received treatment every 7 days with a total of nine doses, the tumors of all mice had completely regressed by Day 90. In the group with treatment every 14 days with a total of five doses, the tumors of two mice were also completely depleted by Day 90, and the tumor volume of the other mice were also greatly reduced, averaging approximately 55 mm^3^ on Day 90. Both treatments had no effect on body weight throughout the experiment, and the survival status of the mice was also normal. For NCI-N87 tumors with an initial tumor volume of 900 mm^3^ and SK-OV-3 tumors with an initial tumor volume of 1100 mm^3^ (Figure 6C,D), we also administered 27 mg/kg Fc-U-Z_HER2_-MMAE (corresponding to 340 nmol/kg MMAE) for treatment once a week for a total of six times. After these treatments, both types of tumors also showed a trend of rapid contraction. On Day 90, the average tumor volume shrank to 82 mm^3^ (NCI-N87) and 245 mm^3^ (SK-OV-3), which were only one-tenth and one-fifth of the initial volumes, respectively. There were no obvious body weight changes, and the physiological status of the mice was also normal in all groups during the whole experiment. These results demonstrated that Fc-U-Z_HER2_-MMAE has a very strong ability to treat large, established NCI-N87 and SK-OV-3 tumors.

The above NCI-N87 and SK-OV-3 tumor-bearing mice treated six times with Fc-U-Z_HER2_-MMAE (27 mg/kg) were dissected, and their serum, tumor tissues and normal organs (heart, liver, spleen, lung and kidney) were collected for blood biochemical analysis and paraffin section staining analysis. H&E and Ki-67 histochemical staining of mouse tumor tissues (Figure 6E) showed that both the NCI-N87 and SK-OV-3 tumor cells in the Fc-U-Z_HER2_-MMAE treatment group showed cytoplasmic defects, increased nucleocytoplasmic ratios and slowed cell proliferation, which were very consistent with the above tumor growth inhibition record. Biochemical analyses of ALT, AST, BUN and CR in the serum (Figure 6F) showed that all of these indicators in all treatment groups were within the normal range, indicating that Fc-U-Z_HER2_-MMAE treatment had no significant negative impact on the liver or kidney functions of mice. H&E staining of the normal mouse organs (Figure 6G), including the heart, liver, spleen, lung and kidney, showed that the samples from the Fc-U-Z_HER2_-MMAE treatment group were consistent with those from the PBS control group. The above results indicated that, after continuous treatment with a high dose of Fc-U-Z_HER2_-MMAE (27 mg/kg, once a week, six times in total), the mice had very good tolerance, and no significant negative effects on normal organs were observed.

### 2.9. Off-Target Toxicity Analysis

We further tested the off-target toxicity of Fc-U-Z_HER2_-MMAE in BALB/c mice. The conjugate was injected into BALB/c mice through tail vein at doses of 23 mg/kg, 45 mg/kg, 90 mg/kg and 180 mg/kg, and the survival status and physiological performance of the mice were recorded over the following two weeks. As shown in Table 1, the mice showed good tolerance to 45 mg/kg and 90 mg/kg Fc-U-Z_HER2_-MMAE. After injection, all mice behaved normally. However, the mice could not tolerate the 180 mg/kg dose. This dosage proved fatal for two of five mice, also causing loss of sight. The maximum tolerance dose (MTD) of Fc-U-Z_HER2_-MMAE in mice may be between 90 and 180 mg/kg. Considering that Fc-U-Z_HER2_-MMAE provided very effective treatment at a dose of 13.5 mg/kg, an MTD of 90–180 mg/kg indicates that Fc-U-Z_HER2_-MMAE may have a broad therapeutic window.

## 3. Materials and Methods

### 3.1. Reagents

Gly3-VC-PAB-MMAE was purchased from Levena Biopharma Co., Ltd. (Cat. No. SET0401, Nanjing, China); 3-(4,5-Dimethylthiazol-2-yl)-2,5-diphenyltetrazolium bromide (MTT, Cat. No. 298-93-1, Beijing, China), 4,6-diamidino-2-phenylindole (DAPI, Cat. No. C0065-10 mL, Beijing, China) and isopropyl-β -D-1-thiogalactopyranoside (IPTG, Cat. No. II0130, Beijing, China) were obtained from Solarbio. The Live/Dead Viability/Cytotoxicity Assay Kit (Cat. No. L6023L) and Hoechst 33342 (Cat. No. H4079) were purchased from US Everbright Inc. (Suzhou, China). The APC Annexin V/PI Apoptosis Detection Kit was purchased from Biolegend (Cat. No. 640932, San Diego, CA, USA). LysoTracker Red DND-99 was purchased from Yisheng Biotechnology Co., Ltd. (Cat. No. 40739ES50, Shanghai, China). HRP-labeled mouse anti-His antibody was obtained from ABclonal (Cat. No. AE028, Wuhan, China). Mouse anti-MMAE antibody and HRP-labeled mouse anti-MMAE antibody were purchased from Hao Xinyuan Biology Company (Cat. No. HXYAE01, Jinan, China). HRP-labeled mouse anti-human IgG Fc antibody was purchased from GenScript (Cat. No. 50B4A9, Suzhou, China). FITC-labeled mouse anti-human IgG Fc antibody was purchased from Proteintech (Cat. No. SA00003-12, Wuhan, China). Human HER2 protein was purchased from ACRO Biosystems (Cat. No. HE2-H5212, Beijing, China). Human urokinase/uPA protein was purchased from Sino Biological (Cat. No. 10815-H08H-A, Beijing, China). RPMI 1640 medium and DMEM were purchased from Gibco BRL (Grand Island, NY, USA). All other materials were obtained from Bestbay Biology Company (Tianjin, China).

### 3.2. Cell Lines and Animals

Human breast cancer BT-474 cells and SK-BR-3 cells were obtained from the American Type Culture Collection (ATCC, Manassas, VA, USA). Human gastric carcinoma NCI-N87 cells, human ovarian cancer SK-OV-3 cells and human breast cancer H-CC70 cells were purchased from the Typical Culture Preservation Commission Cell Bank of the Chinese Academy of Sciences (Shanghai, China). Here, NCI-N87, BT-474 and SK-BR-3 cells had high HER2 expression, SK-OV-3 cells showed mid-level HER2 expression and H-CC70 cells expressed HER2 at a low level. BT-474, SK-BR-3 and NCI-N87 cells were maintained in RPMI-1640 medium, SK-OV-3 cells were maintained in McCoy’s 5A medium, and H-CC70 cells were maintained in DMEM. All media contained L-glutamine and were supplemented with 10% fetal bovine serum (FBS) and 1% penicillin—streptomycin. The cells were incubated in a humidified 37 °C atmosphere containing 5% CO_2_. When adherent cells were 80–90% confluent, a solution of 0.05% trypsin and 0.02% EDTA in phosphate buffer was used for detachment. Human embryonic kidney 293F (HEK-293F) cells were obtained from Abcam (Shanghai, China). Here, HEK-293F cells were used as the expression system, and they were grown in CD293 TEG medium. The cells were maintained in culture at 37 °C with 5% CO_2_ in air and 95% humidity on a 160-rpm shaker.

Female athymic nude mice (BALB/c background) and female BALB/c mice, 6 to 8 weeks old, were purchased from Vital River Laboratory Animal Technology Co., Ltd. (Beijing, China). All mice were housed and treated in accordance with the guidelines of the Committee on Animals of Nankai University (Tianjin, China).

### 3.3. Expression and Purification of the Proteins with LPETGG- 6 × His Tag

The plasmids encoding the proteins Herceptin and Fc-U-Z_HER2_ (human IgG1 Fc) were constructed by inserting the gene sequence into the pFUSE vector (Invitrogen, Carlsbad, CA, USA) with the EcoR I and Nhe I restriction sites. A sortase-recognition motif (LPETGG) and a hexa-histidine tag (6 × His) were added to the C-terminus for drug conjugation. For expression, HEK-293F cells were transiently transfected with the plasmids using PEI (polyethylenimine) as the transfection reagent. The proteins were harvested 7 days post-transfection. Briefly, the cell suspension was centrifuged at 200× *g* for 5 min at 4 °C, and the supernatant was centrifuged again at 5000× *g* for 30 min at 4 °C. The second supernatant was filtered sequentially through 0.45 µm and 0.22 µm PVDF filters and then purified with a protein A column (GE Healthcare, Boston, MA, USA) according to the manufacturer’s instructions. The purity of the obtained proteins was analyzed by 12.5% sodium dodecyl sulfate—polyacrylamide gel electrophoresis (SDS—PAGE).

The plasmid encoding Z_HER2_ was cloned into the pET-28a (+) vector (Novagen, Madison, WI, USA) with Nco I and Xho I restriction sites, with a LPETGG-6 × His tag added to the C-terminus and another 6 × His-tag added to the N terminus for downstream purification of the conjugate Z_HER2_-MMAE. The vector was transformed into BL21 (DE3) pLysS competent cells, and the protein was expressed by induction with 0.5 mM IPTG overnight at 220 rpm and 16 °C. After harvesting by centrifugation (4000 rpm, 20 min, 4 °C), the pellets were resuspended in PBS and disrupted with a high-pressure homogenizer. The crude extract was purified by a nickel chelate affinity chromatography column. The purity of the protein was analyzed by 12.5% SDS—PAGE.

### 3.4. Preparation and Purification of the Conjugates Based on Sortase A

The proteins of Z_HER2_, Herceptin and Fc-U-Z_HER2_ (40 μM) with a LPETGG-6 × His tag added to the C-terminus were mixed with Gly3-VC-PAB-MMAE (400 μM, 10 equiv.) in buffer containing 50 mM Tris-HCl, 150 mM NaCl and 10 mM CaCl_2_ at pH 7.4. Conjugation was initiated by the addition of 2.5 μM sortase A with shaking at 4 °C for 8 h. For Herceptin-MMAE and Fc-U-Z_HER2_-MMAE, the reaction mixtures were purified by a protein A affinity chromatography column and a PD-10 desalting column. For Z_HER2_-MMAE, the reaction mixture was purified by a nickel chelate column and a PD-10 desalting column. The purified protein was analyzed by 12.5% SDS—PAGE and Western blotting using a mouse anti-human IgG monoclonal antibody, a mouse anti-His monoclonal antibody, and a mouse anti-MMAE monoclonal antibody.

### 3.5. Analysis of Binding Affinity Based on ELISA

Enzyme-linked immunosorbent assays (ELISAs) were used for binding ability measurements. Here, 0.1 µg/well of human HER2 tag-free protein (ACROBiosystem, Newark, DE, USA) was added to 96-well plates overnight, followed by blocking for 2 h. After incubation with the conjugates and primitive proteins at a series of concentrations, the mouse anti-MMAE monoclonal antibody as the primary antibody (1 μg/mL) was added to the wells for an additional 1 h of incubation. Diluted HRP-labeled goat anti-mouse antibody (1:5000) in PBS was added for 1 h, and then the samples were incubated with 3,3′,5,5′-tetramethylbenzidine (TMB) for 10 min. After termination with 2 N H_2_SO_4_, the absorbance was measured at 450 nm, and GraphPad Prism was used to calculate the EC_50_ (the concentration needed for 50% of the maximal effect).

### 3.6. In Vitro Cytotoxicity Assays

The in vitro cytotoxicity of the conjugates was measured via an MTT assay. Here, NCI-N87, BT-474, SK-BR-3, SK-OV-3 and H-CC70 cells were seeded at 5.0 × 10^3^ cells/well in 96-well plates and incubated overnight before treatment with various concentrations of conjugates at 37 °C for 72 h. 3-[4,5-Dimethylthiazol-2-yl]-2,5 diphenyl tetrazolium bromide (MTT, 0.5 mg/mL) in PBS was then added, followed by incubation for another 4 h. Then, the supernatant was discarded, the resulting formazan crystals were dissolved in dimethyl sulfoxide (DMSO, 100 μL) with gentle shaking, and the absorbance was measured at 490 nm. IC_50_ values (concentrations at which the cell viability decreased by 50%) were calculated using GraphPad Prism software (GraphPad Software, Inc., San Diego, CA, USA).

### 3.7. In Vivo Pharmacokinetics and Biodistribution

Female BALB/c mice (n = 3) were intravenously injected with a single dose of 5 μg of Z_HER2_-MMAE, Herceptin-MMAE, or Fc-U-Z_HER2_-MMAE in 200 μL of PBS and blood samples were drawn from the ophthalmic vein at different time points (3, 10 and 30 min, 1, 2, 4, 8, 12 and 24 h for Z_HER2_-MMAE; or 3 min, 1, 2, 4, 8, 12, 24, 36, 48, 72, 96, 120 and 196 h for Herceptin-MMAE and Fc-U-Z_HER2_-MMAE). The collected blood samples were maintained at room temperature for 20 min and then centrifuged for separation of the serum. The concentrations of the conjugates to be used were determined with ELISA. Briefly, microtiter plates were coated with human HER2 protein (ACRO biosystems, Cat # HE2-H521y, Beijing, China) in PBS overnight at 4 °C. Then, the plates were blocked with PBS containing 1% bovine serum albumin (BSA) for 2 h at room temperature and washed five times with PBST (PBS with 0.05% Tween-20). The serum samples were diluted to 1:100 in PBS, 100 μL of each sample was added to the appropriate wells, and the plate was incubated for 2 h at room temperature. Mouse anti-His monoclonal antibody or mouse anti-human IgG (Fc) monoclonal antibody was added to the wells as the primary antibody (1 μg/mL) and incubated for an additional 1 h. Diluted HRP-labeled goat anti-mouse antibody (1:5000) in PBS was added and incubated for 1 h, followed by incubation with TMB for 10 min. After termination with 2 N H_2_SO_4_, the absorbance was measured at 450 nm. The absorbance value of each well was normalized to that of the PBS control, and the protein concentrations were quantified using the standard curve. The measured concentrations were plotted using GraphPad Prism software with an exponential decay function to calculate the serum half-life of the conjugates.

The biodistribution of the conjugates was analyzed in a subcutaneous NCI-N87 xenograft tumor model with high HER2 expression. Briefly, 5 × 10^6^ cells were subcutaneously inoculated into the flank region of BALB/c nude mice. After the tumor volumes reached 100–200 mm^3^, the mice were divided into three groups and injected with 20 nmol of the Cy5-labeled conjugates. At 1, 4, 12 and 48 h postadministration, the mice in each group were euthanized, and the tumor grafts and organs (heart, liver, spleen, lung and kidney) were collected. Fluorescence images of the dissected organs and tumors were taken, and the average fluorescence intensity (photons/s/cm^2^/sr) was measured with an IVIS Lumina Imaging system (IVIS Lumina II, Xenogen, Alameda, CA, USA).

### 3.8. In Vivo Antitumor Activity

Subcutaneous xenografts of NCI-N87 tumors with high expression of the HER2 receptor and SK-OV-3 tumors with mid-expression of the HER2 receptor were used to evaluate the in vivo antitumor activity of the conjugates [45,46]. The NCI-N87 tumor model was constructed by subcutaneous inoculation of 5 × 10^6^ NCI-N87 cells per mouse in 100 μL of PBS into the right flanks of the nude mice, and the SK-OV-3 tumor model was constructed by subcutaneous inoculation of 1 × 10^7^ SK-OV-3 cells per mouse in a 1:1 PBS: Matrigel suspension (BD Matrigel, BD Biosciences, San Jose, CA, USA). When tumor volumes reached approximately 100 mm^3^, the mice were randomly divided into groups (n = 5). For treatment, the mice were injected with 170 nmol/kg (calculated based on molar quantity of MMAE) conjugate via the tail vein once a week with a total of six doses, i.e., 26 mg/kg Herceptin-MMAE and 13.5 mg/kg Fc-U-Z_HER2_-MMAE. Considering the short circulation half-life of the conjugate Z_HER2_-MMAE, the administration frequency of this molecule was increased to twice a week for six weeks (3.4 mg/kg) via the tail vein. In the control group, the mice were given PBS at the same time points. Tumor growth was measured with a digital caliper, and tumor volumes were calculated using the following formula: length × (width)^2^/2. The body weights of the animals were recorded throughout the experiment. The animals were euthanized when the tumor volume reached 1500 mm^3^.

For the treatment of large tumors (500–1100 mm^3^), the mice were randomly divided into groups (n = 5) and administered 27 mg/kg (340 nmol/kg based on MMAE) Fc-U-Z_HER2_-MMAE via the tail vein every 7 or 14 days. The changes in tumor volume and animal weight were recorded in the same manner described above.

### 3.9. Detection of Biochemical Indices and H&E and Ki-67 Staining Post-Treatment

Serum was harvested 24 h after the last treatment to examine liver toxicity by measuring ALT (alanine transaminase) and AST (aspartate transaminase) levels and kidney toxicity by measuring BUN (blood urea nitrogen) and Cr (serum creatinine) levels. The biochemical indices were analyzed by an automatic biochemical analyzer (Rayto Chemray 240, Guangzhou, China).

At the end of the experiments, the mice were euthanized, and the tumors and major organs, including the heart, liver, spleen, lung and kidney, were excised to prepare paraffin sections for H&E staining and Ki-67 staining. H&E and Ki-67 staining were performed according to conventional protocols. Images were captured under a microscope (Nikon, Eclipse Ci-L, Tokyo, Japan).

### 3.10. In Vivo Off-Target Toxicity Analysis

BALB/c mice (n = 5) were intravenously injected with a single dose of Fc-U-Z_HER2_-MMAE (23 mg/kg, 45 mg/kg, 90 mg/kg or 180 mg/kg). The physiological changes and mortality of the mice were observed continuously for two weeks postinjection.

### 3.11. Statistical Analysis

Data are presented as the standard error of the mean (SEM), and statistical analysis was performed using one-way ANOVA and Tukey’s significant difference multiple comparisons, * *p* < 0.05, ** *p* < 0.01, *** *p* < 0.001, **** *p* < 0.0001.

## 4. Conclusions

In this study, we proposed a new ADC design strategy of fusion with Fc cleavage by an enzyme, which could simultaneously improve tumor permeability and maintain the long circulation half-life of the drug molecule. In our design, we selected affibody Z_HER2:342_ as the target ligand, whose N-terminal is fused with Fc domain, with a recognition site for the uPA enzyme that is highly active at the tumor site inserted between them, and the sortase A-mediated enzymatic reaction as the site-specifically coupling strategy to construct the target ADC molecules. The experiment results show that, Fc-U-Z_HER2_-MMAE had a good binding ability to HER2 receptor and could kill HER2-positive tumor cells very effectively and selectively at the nanomolar level. Due to Fc fusion and specific uPA cleavage in the tumor site, Fc-U-Z_HER2_-MMAE showed an extended circulation half-life compared to Z_HER2_-MMAE (7.1 h, 59-fold longer) and had significantly enhanced accumulation at the tumor site compared to the conventional full-length antibody-coupled ADC Herceptin-MMAE. These combined effects led to Fc-U-Z_HER2_-MMAE having significantly enhanced tumor treatment ability, as shown in mouse models of NCI-N87 gastric cancer and SK-OV-3 ovarian cancer, where Fc-U-Z_HER2_-MMAE treatment achieved complete regression of tumors in all or a portion of animals with no obvious side effects and an MTD exceeding 90 mg/kg. The above results showed the advantages of this cleavable ADC design strategy for tumor treatment and may provide a new solution for the design of effective ADCs. However, the half-life of Fc-U-Z_HER2_-MMAE was significantly shorter than that of Herceptin-MMAE, maybe due to the presence of the uPA cleavage site. In the future, some new cleavage strategies may need to be attempted in order to improve the stability of the target ADC molecules in the peripheral blood and thus improve circulation half-life.

## Figures and Tables

**Figure 1 ijms-24-11011-f001:**
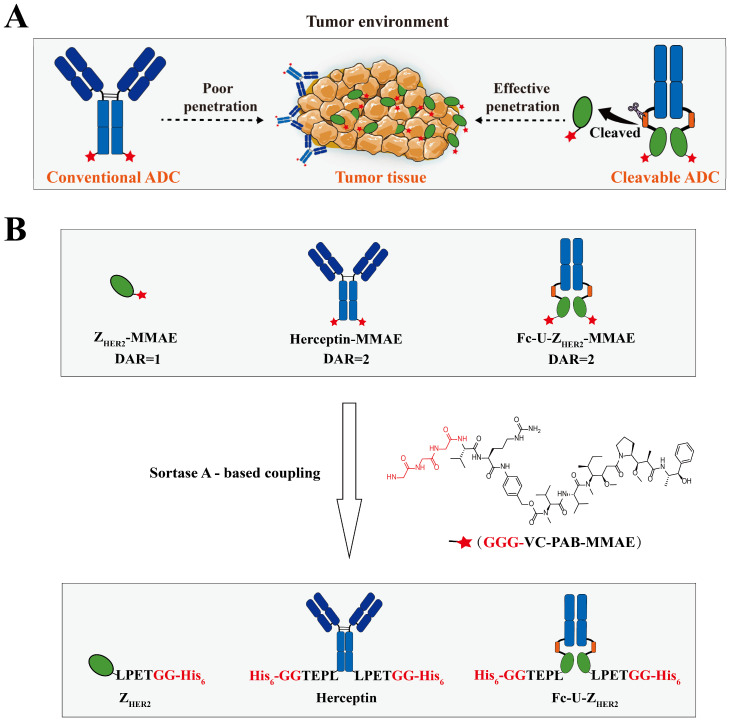
Design and synthesis of the small ADC (Z_HER2_-MMAE), the conventional ADC (Herceptin-MMAE) and the cleavable ADC (Fc-U-Z_HER2_-MMAE). (**A**) Schematic illustration of the general mechanism by which Fc-U-Z_HER2_-MMAE improves tumor treatment through Fc fusion and tumor site-specific cleavage. (**B**) Schematic illustration of the synthesis of Z_HER2_-MMAE, Herceptin-MMAE, and Fc-U-Z_HER2_-MMAE. The raw proteins are Z_HER2_, Herceptin and Fc-U-Z_HER2_, which have one or two LPETGG-His_6_ sequences at the C-terminus. The abbreviation DAR represents the drug-to-antibody ratio.

**Figure 2 ijms-24-11011-f002:**
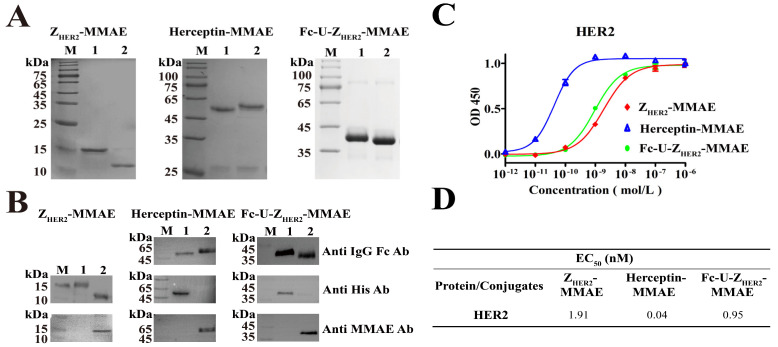
Characterization of the conjugates. (**A**) SDS—PAGE of the purified Z_HER2_-MMAE, Herceptin-MMAE, and Fc-U-Z_HER2_-MMAE. Lane 1, uncoupled raw proteins; Lane 2, conjugates. (**B**) Western blot analysis of Z_HER2_-MMAE, Herceptin-MMAE, and Fc-U-Z_HER2_-MMAE using HRP-labeled mouse anti-IgG Fc antibody, HRP-labeled mouse anti-His antibody, or HRP-labeled mouse anti-MMAE antibody. (**C**) ELISA assays of the binding ability of the conjugates with HER2 receptor. The human HER2 protein was coated on the plates and incubated with different concentrations of Herceptin-MMAE, Z_HER2_-MMAE or Fc-U-Z_HER2_-MMAE at 37 °C for 2 h, and then analyzed with mouse anti-MMAE monoclonal antibody by ELISA. Data are expressed as the mean ± SEM (n = 3). (**D**) EC_50_ data calculated with GraphPad Prism 5.0 software.

**Figure 3 ijms-24-11011-f003:**
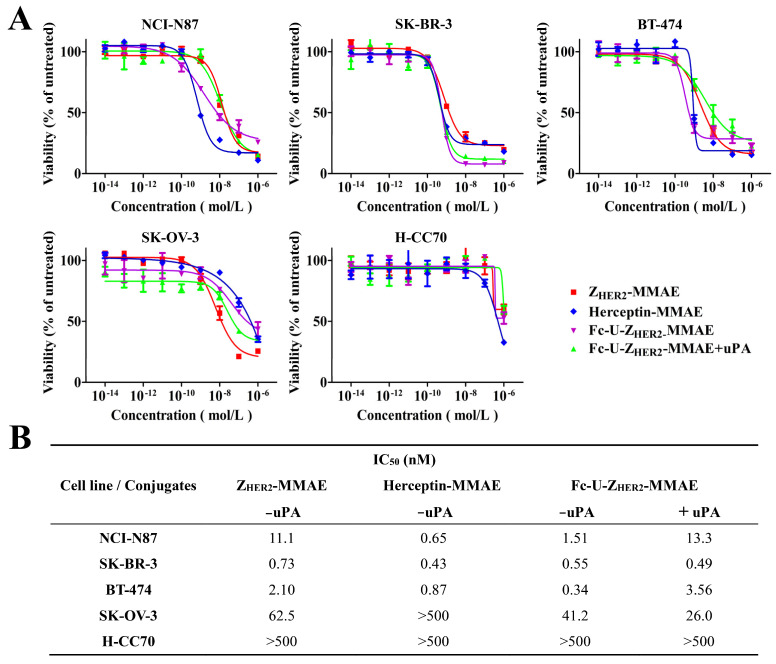
Cell viability determined by MTT assays. (**A**) Cell viabilities after treatment with the conjugates. NCI-N87 cells, SK-BR-3 cells, BT-474 cells, SK-OV-3 cells and H-CC70 cells were incubated with different concentrations of Z_HER2_-MMAE, Herceptin-MMAE and Fc-U-Z_HER2_-MMAE for 72 h, and then the cell viabilities were determined using MTT assays. Data are expressed as the mean ± SEM (n = 3). (**B**) Calculated IC_50_ data.

**Figure 4 ijms-24-11011-f004:**
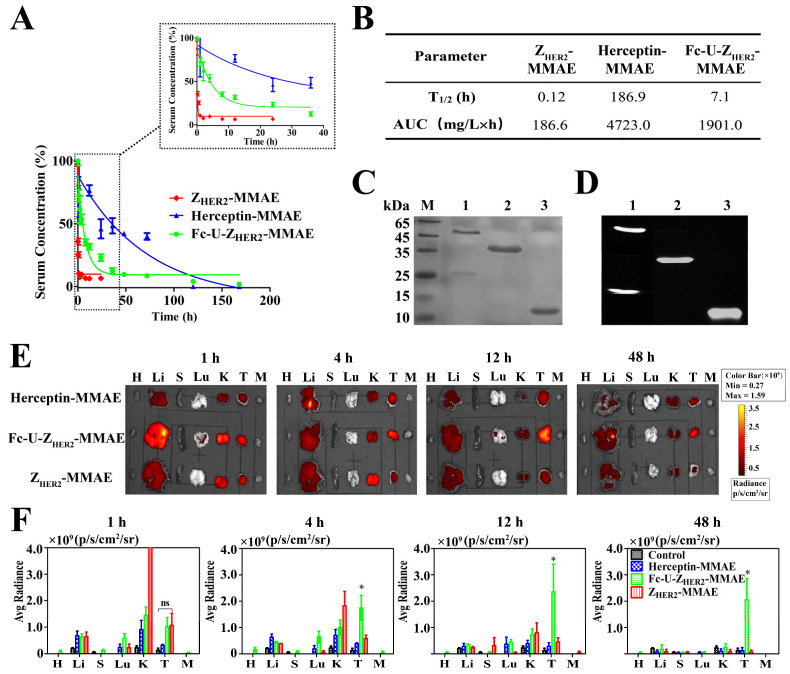
Pharmacokinetics and biodistribution of the conjugates in mice. (**A**) In vivo concentrations of the conjugates in the blood circulation determined by ELISA. The conjugates were injected into the BALB/c mice through a tail vein, and the serum concentrations of the conjugates at different time points were quantified by ELISA. Data are expressed as the mean ± SEM (n = 3). (**B**) Calculated t_1/2_ and AUC values. (**C**,**D**) SDS—PAGE (**C**) and fluorescence imaging (**D**) of the conjugates labeled with Cy5. (**E**) Fluorescence imaging of the in vivo distribution of the conjugates in mice. NCI-N87 tumor model mice were injected with 20 nmol of Cy5-conjugates, and samples were collected at 1, 4, 12 and 48 h for imaging various organs, tumors and muscle tissues. H, heart; Li, liver; S, spleen; Lu, lung; K, kidney; T, tumor; M, muscle tissues. (**F**) Quantified relative average fluorescence intensity. Data are expressed as the mean ± SEM (n = 3). The statistical significance of the differences was analyzed using one-way ANOVA with Tukey’s multiple comparisons with GraphPad Prism 5.0. * *p* < 0.05; ns = not significant.

**Figure 5 ijms-24-11011-f005:**
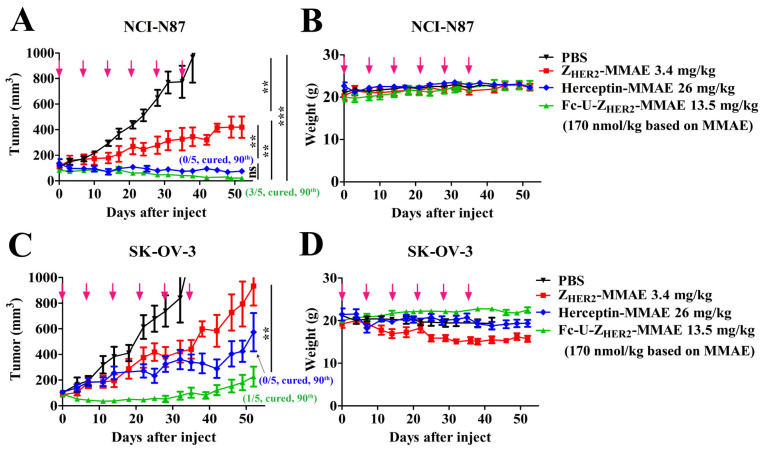
In vivo antitumor activity of the conjugates in mice bearing NCI-N87 and SK-OV-3 tumors. (**A**,**B**) NCI-N87-bearing mice treated with the conjugates. (**C**,**D**) SK-OV-3-bearing mice treated with the conjugates. When tumors grew to 100 mm^3^, the mice were injected with Fc-U-Z_HER2_-MMAE and Herceptin-MMAE at a dosage of 170 nmol/kg based on MMAE on Days 0, 7, 14, 21, 28 and 35 or with Z_HER2_-MMAE twice a week, for a total of twelve times at the same dosage. The changes in tumor sizes (**A**,**C**) and body weights (**B**,**D**) of the mice were recorded. Data are expressed as the mean ± SEM (n = 5). The statistical significance of the differences was analyzed using one-way ANOVA with Tukey’s multiple comparisons with GraphPad Prism 5.0. ** *p* < 0.01, *** *p* < 0.001; ns = not significant.

**Figure 6 ijms-24-11011-f006:**
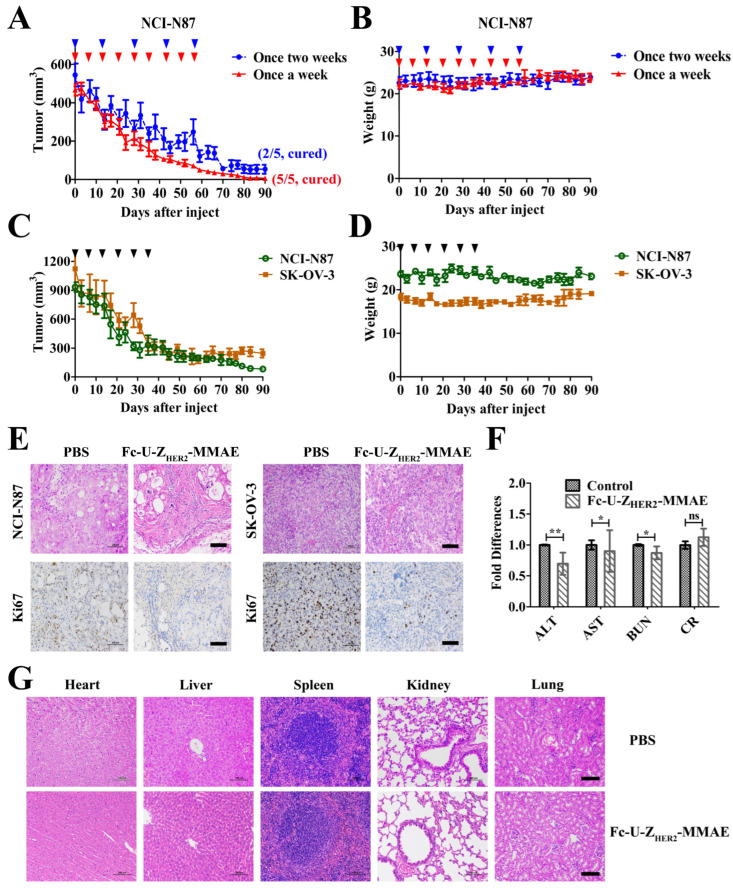
In vivo antitumor activity of Fc-U-Z_HER2_-MMAE in mice bearing large initial tumors. (**A**,**B**) Mice with NCI-N87 tumors with an initial volume of around 500 mm^3^ were treated with Fc-U-Z_HER2_-MMAE. The mice were treated with 27 mg/kg (340 nmol/kg based on MMAE) Fc-U-Z_HER2_-MMAE every 7 or 14 days for nine weeks, and the changes in tumor size (**A**) and body weight (**B**) of the mice were recorded. (**C**,**D**) Mice with NCI-N87 tumors with an initial volume of 900 mm^3^ and with SK-OV-3 tumors with an initial volume of 1100 mm^3^ were treated with Fc-U-Z_HER2_-MMAE. The mice were treated with 27 mg/kg (340 nmol/kg based on MMAE) Fc-U-Z_HER2_-MMAE on Days 0, 7, 14, 21, 28 and 35, and the changes in tumor size (**C**) and body weight (**D**) of the mice were recorded. Data are expressed as the mean ± SEM (n = 5). (**E**) H&E and Ki-67 histochemical staining. After treatment, the tumor tissues of the mice were collected for H&E and Ki-67 immunohistochemical staining to analyze tumor cell proliferation. (**F**) Serum levels of ALT, AST, BUN and CR. (**G**) H&E staining of normal organs (n = 3). Scale bars, 100 µm. Statistical significance was determined by one-way ANOVA with Tukey’s multiple comparisons in GraphPad Prism 5.0 software: * *p* < 0.05, ** *p* < 0.01; ns = no significance.

**Table 1 ijms-24-11011-t001:** Off-target toxicity of Fc-U-Z_HER2_-MMAE (n = 5).

Dead/Total Mice	
Injected Dose (mg/kg)	Fc-U-Z_HER2_-MMAE
23	0/5
45	0/5
90	0/5
180	2/5

## Data Availability

Not applicable.

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
