# Peer review of "Tumor Site-Specific Cleavage Improves the Antitumor Efficacy of Antibody–Drug Conjugates"

_ijms, 2023, doi:10.3390/ijms241311011_

Round 1

Reviewer 1 Report

This is a good manuscript on technically correct and sums up exciting results.

The data from this scientific research supports the therapeutic advantages of the cleavable-ADC strategy, which could provide a new approach for ADC design, both traditional high molecular weight ADCs and small ADCs having limitations and disadvantages in therapy.

A new cleavable ADC design with excellent tumor tissue permeability and a long circulating half-life can provide better tumor penetration. Experimental results support a significantly favorable plasma half-life also.

The manuscript is well-written, presents useful methods for studying the aspects analyzed, and sums up very interesting results.

I suggest some minor revisions:

I have complaints about the layout of Table 1, it is not good quality and doesn`t respect the journal’s requirements, please consider improving it. The same with some figures (Fig. 2, Fig. 3).

The introduction part needs to be rewritten and improved.

The conclusions can be improved with more details, explanations, and the results’ correlations.

Reviewer 2 Report

In this manuscript, three types of conjugated antibody-drug conjugates (ADCs) based on full antibody, affibody-Fc and affibody were synthesized and biologically evaluated. The aim of the paper is engaging and the content is generally well-written. However, I have several concerns that I believe need to be addressed to improve the manuscript. 

His tag can sometimes cause side effects. In my opinion, it should ultimately be removed in drug manufacturing stage. Do the authors have any comments?

I am interested in the developability of this Affibody Fc. Is the titer satisfactory?

The authors should explain why they chose Sortase method as the site-specific conjugation. Also, explanations of their design choices regarding the tag site and DAR are necessary.

The authors should also discuss some of the advances in site-specific ADC manufacturing. Site-specific modification could potentially improve the chemistry, manufacturing and control (CMC) aspects of ADCs and increase the therapeutic index. These points should also be included, and I suggest citing the following references:

For the advantage of ADC analysis by site-selective modification: "Anal. Chem. 2019, 91, 20, 12724-12732"

For the advantage of therapeutic index by site-selective modification: "Nature Biotechnology 2008, 26, 925-932"

Also, the reviewer has some comments regarding biological studies.

Is uPA really involved in therapeutic efficacy? There are doubts. Indeed, in vitro tests in which uPA was co-dosed did not show a significant change in efficacy. If tumor-specific uPA cleavage does occur, was there a need to introduce a Val-Cit moiety in the drug-linker in the first place?

There are also several pharmacokinetic questions. The authors seem to measure only the total antibody concentration. I believe the half-life of trastuzumab in mice is about 15 days, but the half-life of the trastuzumab-uPA antibody in this study appears to be much shorter. The authors suggest that this is due to undesired cleavage at the uPA recognition site, but if that's the case, an accurate antibody concentration should be analyzed by using an anti-Fc antibody as the primary mAb for concentration measurement. Did the authors perform such an experiment?

Did the authors also perform total ADC measurements using an anti-MMAE antibody? It is extremely important in the early evaluation of an ADC to confirm that there is no undesired MMAE detachment.

Why do you perform safety testing in the mouse? Mouse plasma contains carboxylesterases such as Ces1C that are known to cleave at the Val-Cit site. Shouldn't studies be done in rats or preferably monkeys?

The therapeutic efficacy seems weak. For example, the paper below makes a trastuzumab-based ADC with a DAR=2 by site-specific conjugation and shows tumor suppression in an in vivo model at 5mg/kg with 4 doses. The authors' dose of 26 mg/kg in the ADC seems extremely high to me as a reviewer.

Bioconjugate Chem. 2023, 34, 4, 728–738

Also, has dose dependence been confirmed?

Reviewer 3 Report

This manuscript introduces a new cleavable antibody-drug conjugate (ADC) design to target the HER2 receptor. Compared to traditional ADC methods, this new combination shown in this manuscript improved tumor treatment ability in some cancer cells. This manusctipt is well organized and scientifically sound. In addition, considering the importance of target-specific cancer therapies, this study is interesting both in the academic and industrial fields. I suggest several thing to improve the quality of this manuacript.

<Minor points>

1. In Introduction, the authors should explain more specifically why MMAE was chosen as a payload, of many payloads used in ADCs.

2. In 2.1 section (page 2, line 91), the N- and C-terminals of the linker need to be marked.

3. In Figure 1B, the abbreviation "DAR" should be explained as "drug-to-antibody ratio" in the figure legend.

4. What does 342 of Z HER2:342 symbolize? (page 2, line 70) Please explain in the manuscript.

5. As shown In Figure 4B,  the Fc-U-Z-MMAE combination improved a half-life, compared to Z-MMAE. However, it is still shorter than that of Herceptin-MMAE. Thus, the authors can mention any additional techincal strategies to more improve the half-life, in an appropriate section.

6. In Table 1, the dose-serial results are presented. Are there any MTD results of Herceptin-MMAE or Z-MMAE? If there are, the results should be inserted in Table 1.

7. The manuscript showed the promising results of the Fc-U-Z-MMAE combination for target-specific cancer therapies. However, this desing also has technical limitations. Thus, the authors can state these limitations, along with alternatives for future studies in Conclusions section.

Reviewer 4 Report

The manuscript by Keyuan Xu et al., entitled “Tumor Site-Specific Cleavage Improves the Antitumor Efficacy 2 of Antibody–Drug Conjugatespresents an interesting, valuable and well-conducted study on the cleavable ADC design strategy for tumor treatment.

The study is fluently exposed in the manuscript, the materials and methods section is accurately disclosed and the obtained data are relevant and important.

Still, I would like to ask the authors to:

-        Define all the abbreviations used in the text

Mention in the Methods section the number and the date of the Ethics Committee approval.

Round 2

Reviewer 2 Report

The revised manuscript is well-improved. I recommend that it be published as is.